# Preoperative Diagnosis of Warthin Tumors Combining Cytological, Clinical and Ultrasonographic Information within a Multidisciplinary Approach in a Lump Clinic

**DOI:** 10.3390/jpm13071075

**Published:** 2023-06-29

**Authors:** Paolo Fois, Luca Mureddu, Alessandra Manca, Simona Varrucciu, Claudia Crescio, Roberto Gallus, Davide Rizzo, Antonio Cossu, Francesco Bussu

**Affiliations:** 1Division of Otolaryngology, Azienda Ospedaliero Universitaria, 07100 Sassari, Italy; paolo.fois@aouss.it (P.F.); claudia.crescio@aouss.it (C.C.); davide.rizzo@aouss.it (D.R.); fbussu@uniss.it (F.B.); 2U.O.C. Otorinolaringoiatria, Università degli Studi di Cagliari, Policlinico Universitario Duilio Casula, 09042 Monserrato, Italy; 3Institute of Pathology, Azienda Ospedaliero Universitaria, 07100 Sassari, Italy; 4Department of Medicine, Surgery and Pharmacology, University of Sassari, 07100 Sassari, Italy; 5Otolaryngology, Mater Olbia Hospital, 07026 Olbia, Italy

**Keywords:** FNAB, Warthin tumor, FNA, multiparametric analysis, salivary glands, cytology

## Abstract

(1) Background: Warthin tumors account for about 20% of all benign salivary tumors, approaching 50% if we consider only the parotid gland. Wait and see is considered a reasonable option, but the diagnosis should be certain. Diagnosis can be based on morphological and cytological data, but the sensitivity of the fine needle aspiration cytology (FNAC) is not absolute, with a high rate of non-diagnostic findings in the event of a Warthin tumor, hindering the counseling and therapeutic decisions. The aim of the study is to evaluate the reliability of FNAC and its combination with anamnestic, clinical, and ultrasonographic data in diagnosing Warthin tumors. (2) Methods: A total of 413 patients affected by masses within the major salivary gland and managed between 2017 and 2022 at our institution have been included in the present retrospective study. Each patient underwent fine needle aspiration biopsy (FNAB) with a subsequent cytological diagnosis; successively, for each patient, the clinician (otolaryngologist) and the histopathologist discussed the combination of cytological (even non-diagnostic), clinical, and ultrasonographic data in order to make a “multiparametric” diagnosis. A total of 214 cases were subsequently submitted to surgical resection and had a final histopathology report, to which the cytological and the multiparametric diagnoses can be compared. We extracted all the patients with a cytological, multiparametric, and/or histological diagnosis of Warthin tumors in order to assess the sensitivity and specificity of FNAC and of multiparametric analysis in diagnosing Warthin tumors in case of a major salivary gland mass. (3) Results: One hundred thirty-two cases had a cytological, multiparametric, and/or histological diagnosis of Warthin tumors. FNAC displays a sensitivity of 68.4% and a specificity of 98.7% in diagnosing Warthin tumors. The multiparametric evaluation allowed a considerable improvement in sensitivity (92.9% vs. 68.4%), minimizing the number of non-diagnostic results and preserving at the same time a similar value of specificity (95.5% vs. 98.7%). Notably, none of the patients with a cytological or multiparametric diagnosis of Warthin were affected by a malignant lesion in the final histopathological report. (4) Conclusions: In the case of Warthin tumors, a multiparametric evaluation encompassing anamnestic, clinical, and cytological data is effective in reducing the number of non-diagnostic reports and can safely guide the management of a tumor (e.g., antibiotic treatment of infectious complications, assign a low priority to surgery, even consider observation avoiding surgery) which is absolutely benign and can be associated with no clinically relevant issues.

## 1. Introduction

A huge variety of neoplastic and non-neoplastic swellings can be detected within the major salivary glands in humans. Salivary gland tumors (SGTs) are the most common primary neoplasms in these areas, with a wide range of histopathological subtypes that account for approximately 2 to 3% of all tumors of the head and neck region [1]. Most SGTs (80–85%) arising from major salivary glands are benign. Among them, Warthin tumors (WTs) are generally considered the second most common histology (with an incidence ranging from 8 to 30% of all SGTs in different series) after pleomorphic adenoma (64–80%) [2] and involve almost exclusively the parotid gland, representing in some recent studies the most frequent benign neoplasm of this site, even though remarkable epidemiological differences are still present in different series (Table 1).

Smoking, obesity, metabolic syndromes, and HIV infection are known risk factors for WTs [11]. An increased incidence of WTs has been reported in the last few years, possibly due to both an increase in smoking among women and the frequent detection of incidentaloma by advanced imaging techniques [7]. WTs are histologically characterized by bilayered oncocytic epithelium, lymphoid stroma, and cystic spaces filled with viscous turbid material [12] and most commonly present as an asymptomatic, slowly growing mass usually in the fifth and sixth decade with a male-to-female ratio ranging from 2.6:1 to 10.1:1 [13]. WTs are the most common bilateral and multifocal parotid neoplasm. In fact, in about 4–10% of cases, synchronous bilateral WTs can be detected, while in about 4% of the cases, multiple WTs may be observed in the same parotid gland [14]. Differently from pleomorphic adenomas, WTs are characterized by an extremely low potential for malignant transformation (less than 0.1% of the cases) [15], and in turn, they are frequently an incidental finding during oncological follow-up in patients affected by malignant diseases. In fact, WTs have a typical high glucose uptake in PET imaging, which can lead to a misdiagnosis as a distant recurrence [16]. Within the parotid gland, WTs are typically located in the inferior portion of the superficial lobe and can remain stable for years or tend to grow progressively, generally because of recurrent infective events, reaching, in some cases, remarkable dimensions [17].

Magnetic resonance imaging (MRI) is considered by some authors the imaging modality of choice in patients with salivary gland neoplasms, and the presence of certain imaging findings at routine contrast-enhanced MRI may help to differentiate between malignant and benign salivary tumors. However, the imaging findings are often nonspecific [18]. Diffusion-weighted MRI with the calculation of the apparent diffusion coefficient (ADC) is used to differentiate salivary gland neoplasms because tumors with high ADC values are more likely to be benign [19]. Despite this, WTs may contain abundant densely packed lymphoid cells, which accounts for their low ADC value, similar to those of malignant tumors, with misleading findings as those deriving from glucose uptake at PET [20]. Ultrasound (US), and, in particular, surgeon-performed US, is a key resource that could prove helpful in the diagnosis of WTs. The US appearance of such tumors is usually that of an oval or lobulated well-defined hypoechoic lesion, half of the time with a heterogeneous echostructure with cystic areas, mild or no distal acoustic enhancement, and central or mixed vascularization [21].

Cytological evaluation of samples obtained by fine needle aspiration biopsy (FNAB) is a well-established and reliable tool in the diagnostic work-up of head and neck masses without an obvious origin from mucous and/or cutaneous surface, with very low morbidity and no significant risk of tumor seeding [22,23]. It is currently one of the diagnostic steps of the initial evaluation of a neck mass of unknown origin, as recommended by many major international guidelines [24,25,26]. The diagnostic accuracy of fine needle aspiration cytology (FNAC) is drastically increased by US guidance and by the immediate rapid on-site evaluation (ROSE) of the sample by a cytologist with a further sampling of the lesions when deemed necessary [22,27]. Even if surgery currently remains the generally preferred option [28], the clearly benign and sometimes indolent clinical behavior would make active surveillance a valuable option, especially in elderly patients with not-growing long-standing lesions or incidentalomas and no suspicious clinical features [15,29], thus making a reliable diagnosis of WTs potentially extremely useful. The aim of this study is to assess the reliability of US-guided FNAB with ROSE in diagnosing WTs, and whether a multidisciplinary approach integrating such cytological information with clinical and ultrasonographic features can significantly improve FNAC diagnostic accuracy.

## 2. Materials and Methods

We evaluated an institutional database including all patients that underwent a FNAB of head and neck lesions at the Lump Clinic of the Otolaryngology Division of the University Hospital of Sassari, Italy, from September 2017 to December 2022. The study has been conducted according to the principles of the Declaration of Helsinki. Data were analyzed with an observational retrospective design, and in this case, mandatory ethical approval is not requested by Italian law (GU No. 76, 31 March 2008).

Before each FNAB procedure, all previous relevant clinical documentation (CT scan, MRI, ultrasound, PET/CT), the clinical history, and the other relevant information are routinely recorded: time of onset; growth pattern; initial signs and symptoms; personal history of neoplasms or infectious/inflammatory systemic diseases; family history of neoplastic diseases; other clinical conditions; history of smoking habit or alcohol consumption; current medication list. Anamnestic data collection was followed by a physical examination and neck ultrasonography (US) by a Nemio SSA-550A echograph (Toshiba Medical System Corporation, Otawara, Japan) until January 2021 and then by an Acuson NX3 Elite (Siemens Healthcare s.r.l., Erlangen, Germany) coupled with a 12 MHz 30 mm linear probe. The physical examination always included inspection and palpation of the neck surface (including the skin) and the assessment of the characteristics of the neck mass, including approximative location, size, shape, mobility, consistency (soft vs. firm), and ulceration of overlying skin. Upper airway endoscopy and palpation of the mucosal surfaces were also performed as needed and indicated. The ultrasonographic parameters of the lesions routinely recorded included: side of the lesion; anatomical region (i.e., face, lateral neck, median neck, or oropharynx-oral cavity); site (i.e., parotid, submandibular gland, lymph nodal levels of the neck, thyroid, oropharynx, lip, cheek, temporo-zygomatic region or other median neck); largest diameter/s; overall echogenicity (i.e., hyper-, hypo-, iso-, or anechoic; shape of the lesion (i.e., round, ovoid, polycyclic, lobulated or irregular); contour (smooth, irregular); margins (well-/ill-defined); texture (homogeneous, inhomogeneous); cystic component (present, absent); distal phenomena (posterior acoustic enhancement or shadowing); vascularity (central, peripheral); sonographic palpation (compressible or not); and presumable possible origin.

A team of otolaryngologists and histopathologists performed all the FNA procedures as previously described [22]. In brief, our standard procedure includes local disinfection with povidone–iodine 10% solution (Betadine; Purdue Frederick Co., Stamford, Connecticut) and placement of the US probe on the lesion with the major axis of the probe along the desired path of the needle that should be visible at all times as the exact target within the lesion. Aspiration biopsies are performed by means of a 27-gauge needle attached to a 20 mL syringe mounted on a Cameco pistol. The general characteristics of the aspirated material are routinely recorded, including aspect (serous, purulent, hematic) and quantity. All specimens are immediately stained with the May Grunwald Giemsa-fast method kit (DiaPath s.p.a., Martinengo, Italy), and a subsequent rapid on-site evaluation (ROSE) is always performed by an expert cytopathologist. If the material is deemed inadequate or insufficient, up to three aspirations are performed. The collected specimens are then smeared on a microscope slide, alcohol-fixed, stained with hematoxylin and eosin, and analyzed by an expert cytopathologist of the Department of Anatomic Pathology, according to standard guidelines. Since 2019 cytologic reports of salivary gland lesions routinely include classification according to the Milan System for Reporting Salivary Gland Cytopathology (MSRSGC): I: non-diagnostic; II: non-neoplastic; III: atypia of undetermined significance (AUS); IVa: neoplasm-benign; IVb: neoplasm-salivary gland neoplasm of uncertain malignant potential (SUMP); V: suspicious of malignancy; VI: malignant [30]. All the cases are then reevaluated by the otolaryngologist and the histopathologist, combining the cytological findings with clinical history, physical examination, ultrasonographic features, and further imaging data, if any, and a formal consensus is reached on a diagnosis we will refer to as multiparametric diagnosis before any clinical decision, in particular about surgery, is taken. FNA findings not pathognomonic but compatible with WTs to the aim of multiparametric diagnosis have been considered the following: abundant histiocytes and/or neutrophils, and/or the presence of normal lymphocytes [31] as well as the aspiration of abundant (>1 mL) fluid, serous, or purulent material.

For the aim of the present work, we selectively analyzed cases with a cytological, multiparametric, and/or histological diagnosis of WTs in order to assess the reliability of the two assays in diagnosing WTs by computing their specificity and sensitivity. Statistical analysis was performed using JMP software, release 7.0.1, from the SAS Institute.

## 3. Results

Data collection from our database retrieved a total of 633 FNAB reports. A total of 413/633 (65.2%) FNAB were performed on major salivary gland lesions. In 128 out of 413 (31%) FNAB on major salivary glands, the multiparametric analysis suggested a diagnosis of WT, while 92/413 (22.2%) cytology reports were consistent with WTs. In 214/413 (51.8%) major salivary gland cases, surgery was performed, and a definitive histological report was available. Among 214 histopathology reports, 57 (26.6%) were consistent with a diagnosis of WT, while 41 FNAB (19.16%) and 60 multiparametric analyses (28%) were positive in the same subgroup. Contingency tables showing the results of FNACs and multiparametric analysis compared to histology reports are shown in Table 2. Specificity and sensitivity of cytology were, respectively, 98.7% and 68.4%, while specificity and sensitivity of multiparametric analysis were, respectively, 95.5% and 92.9%. Among the 57 histologically confirmed WTs, 15 (26%) had a “non-diagnostic” FNAC report, and only one had a non-diagnostic multiparametric report. We decided to group the non-diagnostic FNAC reports with the negative “ones” as in both, there was not a diagnosis of WT.

Therefore, the negative predictive value (NPV) was noticeably improved with multiparametric analysis compared to that of cytology alone (97.4% vs. 89.6%), while the positive predictive value (PPV) was worse with multiparametric (88.3% vs. 95.1%), mainly because of the increased number of false positive results (seven vs. two cases). Nevertheless, none of these patients had a final diagnosis of malignancy (one pleomorphic adenoma, two oncocytomas, one myoepithelioma, and three non-neoplastic/inflammatory lesions). Diagnostic accuracy was 90.6% for FNAB cytology and 94.9% for multiparametric analysis.

Overall, WT cases suggested by FNAC, multiparametric analysis, or histopathology were 132, and 64 of these (48.5%) had a histologic specimen coming from surgical treatment, consistent with WTs in 57 cases (Figure 1).

The main US features of these 132 cases are reported in Table 3. Briefly, in most of these cases, the US showed hypoechoic lesions (66, 50%), compressible (31, 23.5%), with an oval shape (74, 56.1%), smooth contours (42, 31.8%), sharp margins (78, 59.1%), homogeneous texture (65, 49.2%), no cystic component (55, 42%), no fluid content (93, 70.5%), and distal acoustic enhancement (70, 53%), with a mean diameter of 24.18 mm (std dev +/− 12.6 mm). Most (97%) of these cases were correctly identified as a salivary lesion with a Milan classification IVa in 76.7% of cases (92/120 available), IVb in 0.8% (1/120), and I in 22.5% of cases (27/120).

In total, 92/132 (69.7%) of these cases had a cytology report consistent with WTs, while 128/132 (97%) were consistent with WTs, according to the multiparametric analysis. Complete FNAB and multiparametric analysis results for these cases are shown in Figure 2, while the results for the 57 cases with a final histopathologic report consistent with WTs can be seen in Figure 3. None of the patients with a cytological or multiparametric diagnosis of WT for whom the histological report was available had a final diagnosis of malignant neoplasm. All of the 68 cases that were not finally surgically treated had a multiparametric diagnosis of WT, but only 51/68 (75%) were WTs, according only to the cytopathology report. The operated patients were, on average, younger than the non-operated ones (63 vs. 66 years, *p* = 0.1 at *t*-test), and the operated lesions were significantly larger than the non-operated ones according to the US performed by the surgeon before the FNAB (28.6 mm vs. 20.2 mm on average in their larger diameter, *p* = 0.0002 at *t*-test). Most of the tumors (130, 98.5%) were located within the superficial lobe of the parotid gland, except two (1.5%) located within the submandibular gland.

## 4. Discussion

WTs are benign lesions with a negligible rate of malignant transformation. The reasons why they are first noticed as neck masses and studied according to international guidelines [24,25,26] are usually the following:Growth, usually above 2 cm, as smaller lesions are clinically barely detectable also because of their soft consistency;Recurrent infections, usually associated with an enlargement of the mass [17]. In some cases, they can be acute and prominent: in our experience, most of the unilateral acute swellings and infections of the parotid are infected WTs. In these cases, after medical therapy and fine or gross needle aspiration of purulent content, with the resolution of the acute inflammation, a cystic mass remains in the gland without completely disappearing. When a parotidectomy is subsequently performed, the clinical diagnosis of WT is usually histologically confirmed;Occasional finding (incidentaloma) after a head and neck region imaging [7], often from a positive PET/CT scan performed during oncological follow-up. The latter situation can lead to the misdiagnosis of distant relapse and create relevant psychological distress for the patient.

Resection of these lesions has a lower priority than most of the other neoplasms arising within major salivary glands, and active surveillance would be a valuable option under certain conditions. Among such conditions, there is an elderly age, a low performance status, comorbidities, substantial absence of growth, no clear acute infections, no cosmetic relevance, patient preference, but most of all, a reliable diagnosis of WTs. Conversely, relevant tumor growth, clinical problems, and above all, an uncertain diagnosis are clear surgical indications [28].

In the present series, none of the lesions diagnosed as a WT at FNAB or multiparametric diagnosis turned out to be malignant, thus supporting the described diagnostic approach, which appears safe.

Out of the 132 lesions with a diagnosis of WT, 68 (all diagnosed by FNAB and/or multiparametric analysis) were not surgically treated in the present series, with a considerable reduction in morbidity mainly allowed by a reliable diagnosis. Notably, the non-operated patients were older, and the non-resected lesions were significantly smaller, consistently with the above-cited criteria considered when defining the recommendation.

However, a non-diagnostic result, as it was in 25% of the cytopathology reports of the non-operated lesions, would not have allowed to safely recommend active surveillance with avoidance of surgery, and consequent risks, morbidities, and costs. In these cases, our multiparametric approach has proven to be very useful. Multiparametric diagnosis is ultimately a consensus between the cytopathologist and surgeon based upon the combination of cytological findings, non-diagnostic ones in particular, and clinical and ultrasonographic parameters. Cytological findings in the smear not pathognomonic but compatible with WTs are abundant histiocytes and/or neutrophils and/or the presence of normal lymphocytes [31]. Another common finding in WTs, especially those without clearly diagnostic cytological findings, is the aspiration of abundant (>1 mL) fluid, serous, or purulent material. Clinically, WTs are usually soft at palpation. The clinical onset/pattern of growth reported by the patients can be different, ranging from stability/slow growth to volumetric variations with a relapsing–remitting behavior, with enlargements possibly concomitant with superinfections, and partial regressions, often induced by medical treatment, to the total absence of clinical manifestations until the occasional detection (incidentaloma). Facial paralysis is not usually associated with WTs, a rare exception being an aggressive infectious complication where it usually recovers upon regression of the acute infection after medical treatment. Ultrasound too proves useful, often showing the classical round-oval regular shape, sometimes with a characteristic cystic component, fluid content, and central/peripheral vascularization, a location in the inferior pole of the gland and/or superficial to the external/anterior aspect of the sternocleidomastoid muscle (Figure 4).

In the present analysis, we decided to group the non-diagnostic reports together with the negative ones, as in both cases, there was not a cytological diagnosis of WT, with the aim to obtain a realistic estimate of the diagnostic power. With this premise, our board, evaluating clinical, cytological, and US data, performed better than FNAC alone, reaching a specificity and sensitivity, respectively, of 95.5% and 92.9% as compared to FNAC’s 98.7% and 68.4%. Notably, other authors reported issues concerning the sensitivity of FNAC in the diagnosis of WTs, with figures often below 70%, even when non-diagnostic reports were not considered in the estimate [15,32,33,34]. Our data show that, in the case of a non-diagnostic FNAC report but with cytological findings compatible with WTs, the combination of these with the clinical and ultrasonographic information can lead to a reliable diagnosis of WT. Thus, with such a multiparametric approach, it appears possible to considerably improve accuracy in identifying preoperatively WTs, mainly reducing the non-diagnostic results of FNAB, without increasing concurrently the risk of missing malignant lesions and allowing safe, conservative management of such lesions in selected cases. As it is, the described multiparametric approach is potentially heavily affected by a notable degree of subjectivity, yet it could be the conceptual ground to build a mixed score, built upon quantitative or semiquantitative clinical, ultrasonographic, and cytological information, able to appropriately identify patients affected by WTs even in case of non-diagnostic, yet still compatible, FNAC findings.

## 5. Conclusions

Although benign, WTs represent a challenge for the surgeon and for the patient, inherently and precisely because of their benign nature and the undesirable potential complications resulting from an unnecessary sialoadenectomy. While one could be tempted to rely on FNAB’s results alone or prevalently to characterize the lesion and counsel the patient, such an attitude could lead to a certain amount of misdiagnosed WTs and sometimes to unnecessary surgical treatments. Our approach, albeit still simple and not organized in an objective score, is able to correctly identify WTs as such even when the FNAB misses the diagnosis, leading to more precise and efficient counseling to the patients, especially those unwilling or not fit for a surgical treatment.

## Figures and Tables

**Figure 1 jpm-13-01075-f001:**
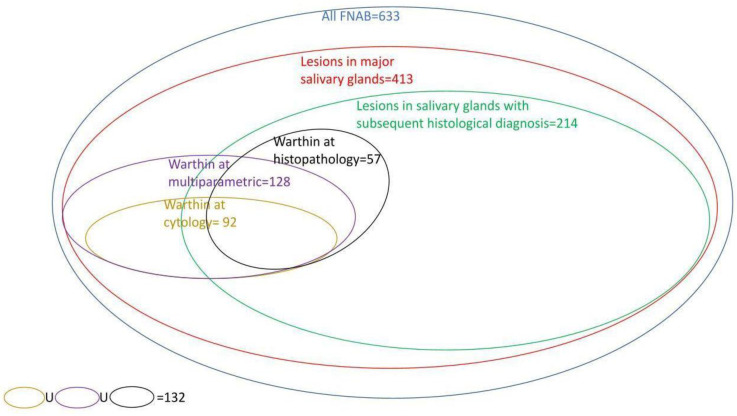
Graphical representation of the diagnostic subgroups within the total number of cases analyzed.

**Figure 2 jpm-13-01075-f002:**
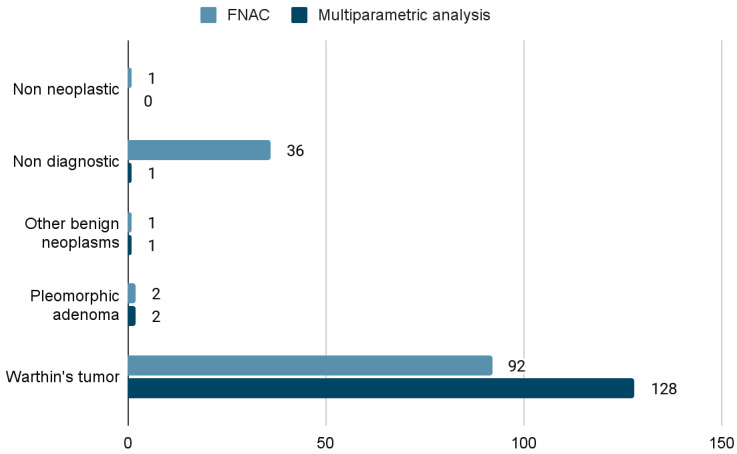
Fine needle aspiration cytology (FNAC) and multiparametric analysis results for the subgroup of patients with possible or definite WTs (cytology + multiparametric + histology, 132 cases).

**Figure 3 jpm-13-01075-f003:**
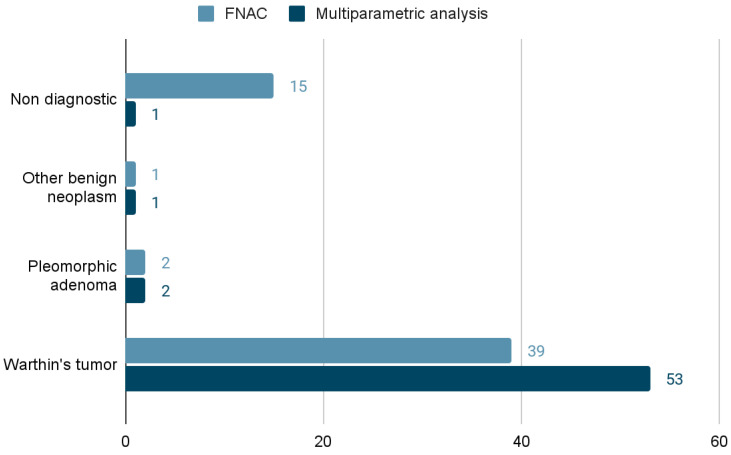
FNAC and multiparametric analysis results for the subgroup of patients with definite (histology, 57 cases) WTs.

**Figure 4 jpm-13-01075-f004:**
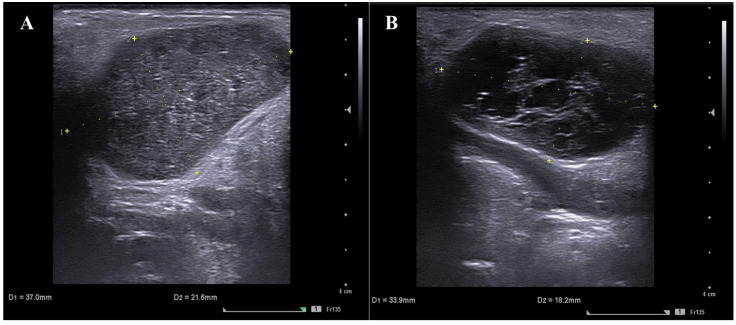
Typical US findings in WTs of the parotid gland: oval shape lesion with smooth contour, sharp margins, and mild distal acoustic enhancement (**A**), in a more superficial position in relation to the external surface of the sternocleidomastoid muscle and with an internal cystic component (**B**).

**Table 1 jpm-13-01075-t001:** Epidemiology of benign salivary glands tumors (SGTs): prevalence of Warthin tumors (WTs) and pleomorphic adenoma (PA) in recent series.

First Author, Year of Publication (Origin)	N. of Cases	% WT of Benign SGT	% PA of Benign SGT	% WT of Benign Parotid Tumors	% PA of Benign Parotid Tumors
Alsanie, 2022 (International) [2]	3751	17%	70%		
Jaremek-Ochniak 2022 (Poland) [3]	336	36%	54.10%	37.50%	52.70%
Al-Balas, 2021 (USA-Jordan) [4]	82			54.90%	18.30%
Ayral, 2021 (Turkey) [5]	108			32.40%	60.20%
Galdirs, 2021 (Germany) [6]	307	9.4%	90.2%		
Psychogios, 2020 (Germany–Greece) [7]	474			42.40%	29.10%
Reinheimer, 2019 (Brazil) [8]	124	4.90%	91.30%		
Franzen, 2018 (Germany) [9]	629			42.1%	35.4%
Fiorella, 2005 (Italy) [10]	4718			32.40%	57.30%

**Table 2 jpm-13-01075-t002:** Fine needle aspiration cytology (FNAC) and multiparametric analysis results compared to histological results in the operated group (total of 214 patients). Source data for sensitivity, specificity, and accuracy analysis. TP: true positive, TN: true negative, FP: false positive, FN: false negative, PPV: positive predictive value, NPV: negative predictive value.

FNAC and Histology Results				
	Histology WT+	Histology WT−	Total	
FNAC WT+	39 (TP)	2 (FP)	41	PPV: 95.1%
FNAC WT−	18 (FN)	155 (TN)	173	NPV: 89.6%
Total	57	157	214	
	Sensitivity: 68.4%	Specificity: 98.7%		Accuracy: 90.6%
Multiparametric Analysis and Histology Results				
Multiparametric WT+	53 (TP)	7 (FP)	60	PPV: 88.3%
Multiparametric WT−	4 (FN)	150 (TN)	154	NPV: 97.4%
Total	57	157	214	
	Sensitivity: 92.8%	Specificity: 95.5%		Accuracy: 94.9%

**Table 3 jpm-13-01075-t003:** Ultrasound (US) characteristics of the 132 possible and definite cases of WTs.

Ultrasound Characteristics							
Echogenicity	Isoechoic 4 (3%)	Hypo/Isoechoic 27 (20.5%)	Hypoechoic 66 (50%)	Iso/Anechoic 1 (0.8%)	Hypo/Anechoic 27 (20.5%)	Anechoic 3 (2.3%)	Not Reported 4 (3%)
Shape	Spindle- shaped 1 (0.8%)	Round 31 (23.5%)	Polyciclic/ Lobulated 5 (3.8%)	Oval 74 (56.1%)	-	-	Not reported 21 (15.9%)
Contour	Irregular 4 (3%)	Smooth 42 (31.8%)	-	-	-	-	Not reported 86 (65.2%)
Margins	Distinct/ Sharp 78 (59.1%)	Ill-defined 6 (4.5%)	-	-	-	-	Not reported 48 (36.4%)
Texture	Homogeneous 65 (49.2%)	Inhomogeneous 28 (21.2%)	-	-	-	-	Not reported 39 (29.5%)
Cystic component	Yes 11 (8.4%)	No 55 (42%)	-	-	-	-	Not reported 65 (49.6%)
Fluid content	None 93 (70.5%)	Purulent 19 (14.4%)	Serous 14 (10.6%)				Not reported 6 (4.5%)
Distal Phenomena	Distal acoustic enhancement 70 (53%)	None 58 (43.9%)	-	-	-	-	Not reported 4 (3%)
Sonographic Palpation Compressible	Yes 31 (23.5%)	No 9 (6.8%)	-	-	-	-	Not reported 92 (69.7%)
Presumable origin	Dysmorphic 2 (1.5%)	Lymph node 1 (0.8%)	Other 1 (0.8%)	Salivary 127 (96.2%)	-	-	Not reported 1 (0.8%)
Diameter	75% quartile 30.53 mm	50% median 22 mm	25% quartile 15 mm	Mean 24.18 mm	Std Dev 12.60	-	-

## Data Availability

Original data are available upon reasonable request.

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
