# Peer review of "Preoperative Diagnosis of Warthin Tumors Combining Cytological, Clinical and Ultrasonographic Information within a Multidisciplinary Approach in a Lump Clinic"

_jpm, 2023, doi:10.3390/jpm13071075_

Round 1

Reviewer 1 Report

While the study is somewhat interesting, I don’t believe these findings add anything to the literature. Any diagnostic test should be placed in the context of the clinical and imaging findings. Also, the diagnostic role of and high accuracy of FNA not only for salivary gland lesions overall but for WT in particular is not accurately reflected in this study. The sensitivity and specificity of FNA for WT is usually very high, so some of the arguments made in this study are not based on the literature as a whole (but rather selected references and interpretations).

Aside from this, the methods are not well explained or robust. For instance, the exact features and systematic approach to placing lesions to a “consistent with WT” category based on the multiparametric approach are not well explained which makes drawing any conclusions challenging. Some of the conclusions are also not appropriate as they are not drawn from appropriate comparisons or based in strongly supported data.

Overall, this study would greatly benefit from input from members of your cytopathology team.

Sufficient

Author Response

Comments and Suggestions for Authors
While the study is somewhat interesting, I don’t believe these findings add anything to the literature. Any diagnostic test should be placed in the context of the clinical and imaging findings.
-    Thank you very much for your precious observations. Actually, placing the FNAC in the context of the clinical and imaging findings in a systematic way is the intent of our effort, we tried to better explain it in the text in the new version of the manuscript, by modifying the following sentence. “All the cases are then reevaluated by the otolaryngologist and the histopathologist together, combining the cytological findings with clinical history, physical examination, ultrasonographic features and further imaging data, if any, and a formal consensus is reached on a diagnosis we will refer to as multiparametric diagnosis before any clinical decision, in particular about surgery, is taken.”

 Also, the diagnostic role of and high accuracy of FNA not only for salivary gland lesions overall but for WT in particular is not accurately reflected in this study. The sensitivity and specificity of FNA for WT is usually very high, so some of the arguments made in this study are not based on the literature as a whole (but rather selected references and interpretations).
-    Thank you very much for the observation. We completely agree with the reviewer about the role and accuracy of FNA and we are strong supporters of its widespread use in otolaryngology, especially in the approach to head and neck masses. However, in our experience FNA can lack sensitivity in diagnosing WTs probably because they often come to observation for infectious complications, and in these cases it can be very difficult to make a purely cytological diagnosis. However, we may have missed some very relevant previous paper on the matter and we would be very grateful if the reviewer could help us to amend our mistake by providing such references.

Aside from this, the methods are not well explained or robust. For instance, the exact features and systematic approach to placing lesions to a “consistent with WT” category based on the multiparametric approach are not well explained which makes drawing any conclusions challenging. 
-    Thank you very much for the observation, it is clearly a fundamental aspect of the work and needed to be better clarified. To this aim we added the following sentence in the material and methods section: “FNA findings not pathognomonic but compatible with WT to the aim of multiparametric diagnosis have been considered the following: abundant histiocytes and/or neutrophils, and/or the presence of normal lymphocytes [31] as well as the aspiration of abundant (>1 ml) fluid, serous or purulent, material”. Other details about clinical history, ultrasonographic and clinical features are provided in the discussion section. We agree that it remains a subjective approach, however we plan to make such an approach more objective by validating scores/nomograms/algorithms as stated in the conclusions.

Some of the conclusions are also not appropriate as they are not drawn from appropriate comparisons or based in strongly supported data.
-    Thank you for the observation, it is a preliminary work demonstrating that our multiparametric diagnosis, evaluating clinical, cytological, and US data, performed better than cytology alone, reaching a specificity and sensitivity respectively of 95,5% and 92,9% as compared to FNAC’s 98,7% and 68,4%. We think that, even if still unable to change the daily clinical practice, it can be a good starting point to improve the clinical approach to cervical masses and WT in particular.

Overall, this study would greatly benefit from input from members of your cytopathology team.
-    Thank you for your suggestion. Our cytopathology team (Dr. Alessandra Manca and Prof Antonio Cossu) was involved at all phases during the conception and writing of the article, and has been involved during the revision process.

Comments on the Quality of English Language
Sufficient

Reviewer 2 Report

The work deals with a benign process of the sophoid glands - Whartin's tumor. As a result, a multi-parametric tool is presented, which combines FNAB and numerous clinical parameters.

The diagnostic accuracy is very high, malingomas were not overlooked 

Possibly a recent German paper by Galdirs et al (PMID 33429442) would complement Table 1 very well from a current perspective. 

Author Response

Comments and Suggestions for Authors

The work deals with a benign process of the sophoid glands - Whartin's tumor. As a result, a multi-parametric tool is presented, which combines FNAB and numerous clinical parameters.

The diagnostic accuracy is very high, malingomas were not overlooked

Possibly a recent German paper by Galdirs et al (PMID 33429442) would complement Table 1 very well from a current perspective.

  • Thank you, we added the paper in table 1, unfortunately we overlooked it because our research team focused only on papers written in english. Actually it helps us to better highlight the remarkable dissimilarity of warthin tumors and pleomorphic adenomas prevalence shown in different recent series.

Reviewer 3 Report

It's a well-written manuscript that evaluates the Warthing Tumors diagnoses accuracy under the FNAB technique with or without multiparametric analysis.

Introduction: adequate;

Materials and Methods: The authors present the methodology in a clear way that favours its reproduction.

 - Although the work took place in a national institution in Italy, which, according to local legislation, does not require the evaluation of an ethics committee, the authors must declare that the present study is by the Declaration of Helsinki because his study uses secondary data that allow the identification of those involved throughout the research analyses.

Results: The results are clear and objective.

Discussion: Adequate.

Conclusion: Adequate.

Author Response

Comments and Suggestions for Authors

It's a well-written manuscript that evaluates the Warthing Tumors diagnoses accuracy under the FNAB technique with or without multiparametric analysis.

Introduction: adequate;

Materials and Methods: The authors present the methodology in a clear way that favours its reproduction.

Although the work took place in a national institution in Italy, which, according to local legislation, does not require the evaluation of an ethics committee, the authors must declare that the present study is by the Declaration of Helsinki because his study uses secondary data that allow the identification of those involved throughout the research analyses.

  • Thank you, we added the statement in the material and methods section.

Results: The results are clear and objective.

Discussion: Adequate.

Conclusion: Adequate.